# Heterologous Expression of Human Metallothionein Gene *HsMT1L* Can Enhance the Tolerance of Tobacco (*Nicotiana nudicaulis* Watson) to Zinc and Cadmium

**DOI:** 10.3390/genes13122413

**Published:** 2022-12-19

**Authors:** Yilin Zheng, Meng Cui, Lei Ni, Yafei Qin, Jinhua Li, Yu Pan, Xingguo Zhang

**Affiliations:** 1Key Laboratory of Horticulture Science for Southern Mountainous Regions, the Ministry of Education, College of Horticulture and Landscape Architecture, Southwest University, Beibei, Chongqing 400715, China; 2State Cultivation Base of Crop Stress Biology for Southern Mountainous Land of Southwest University, Academy of Agricultural Sciences, Southwest University, Beibei, Chongqing 400715, China

**Keywords:** human metallothionein 1L gene, tobacco, zinc, cadmium, heavy metal pollution

## Abstract

Metallothionein (MT) is a multifunctional inducible protein in animals, plants, and microorganisms. MT is rich in cysteine residues (10−30%), can combine with metal ions, has a low molecular weight, and plays an essential biological role in various stages of the growth and development of organisms. Due to its strong ability to bind metal ions and scavenge free radicals, metallothionein has been used in medicine, health care, and other areas. Zinc is essential for plant growth, but excessive zinc (Zn) is bound to poison plants, and cadmium (Cd) is a significant environmental pollutant. A high concentration of cadmium can significantly affect the growth and development of plants and even lead to plant death. In this study, the human metallothionein gene *HsMT1L* under the control of the CaMV 35S constitutive promoter was transformed into tobacco, and the tolerance and accumulation capacity of transgenic tobacco plants to Zn and Cd were explored. The results showed that the high-level expression of *HsMT1L* in tobacco could significantly enhance the accumulation of Zn^2+^ and Cd^2+^ in both the aboveground parts and the roots compared to wild-type tobacco plants and conferred a greater tolerance to Zn and Cd in transgenic tobacco. Subcellular localization showed that *HsMT1L* was localized to the nucleus and cytoplasm in the tobacco. Our study suggests that *HsMT1L* can be used for the phytoremediation of soil for heavy metal removal.

## 1. Introduction

With the development of modern industry and agriculture, heavy metal pollution has seriously affected plant and human life [1,2,3,4]. Heavy metals can cause toxicity to plants by increasing the production of reactive oxygen species (ROS) such as superoxide anion radicals (O^2−^), hydrogen peroxide (H_2_O_2_), and hydroxyl radicals (-OH) [5,6]. Due to crop uptake and accumulation, toxic heavy metals can enter the food chain, posing a threat to human health [7,8]. Conventional physical and chemical methods for removing heavy metals from polluted environments are usually not widely used and are typically costly [9]. Phytoremediation is a new and effective method for removing heavy metal pollution from the soil. However, some problems exist, such as the slow growth, low biomass, and limited enrichment ability of common plants [10,11,12].

Plant genetic engineering technology can transfer genes for heavy metal accumulation or enrichment into plants [13]. Metallothionein (MT) is a multifunctional inducible protein widely present in animals, plants, and microorganisms [14]. MT polypeptide chains generally have 61–62 amino acid residues, and cysteine content accounts for one-third of the total amino acids [15]. All cysteine thiols are coordinated with metal ions to form metal-mercapto clusters, which can strongly chelate toxic metals [16]. Metallic species of zinc, cadmium, mercury, and copper bind to MT in clusters [17]. MT is superior to other antioxidants in scavenging free radicals due to its reduced thiol group. Its ability to scavenge hydroxyl radicals (-OH) is about 10,000 times that of superoxide dismutase (SOD), and its ability to scavenge oxygen radicals (O_2_^−^) is about 25 times that of GSH [18].

According to the differences in amino acid composition and isoelectric points, mammalian metallothionein can be divided into four isomers: MT1, MT2, MT3, and MT4 [19,20]. Among them, MT1 and MT2 are the primary heterogeneous forms, widely present in almost every tissue to maintain homeostasis [21,22]. Plant MT can be divided into three categories according to the position and arrangement of its cysteine residues. The Cys of type I plant MT is arranged in a C-C, C-X-C, and C-X-X-C manner, with a concentrated distribution in the N-terminal and C-terminal of the peptide chain, and is also the most commonly found class. The Cys of type II is dispersed throughout the entire protein sequence. Type III is not a product of gene coding, and it is mainly a polymer synthesized by glutathione catalyzed by plant chelating peptide synthase, also known as phytochelatins [23].

Many studies have confirmed the detoxification function of metallothionein for heavy metals. Among all known metallothionein types, mammalian MT1 and MT2 have the strongest metal binding capacities and have been extensively investigated in recent decades [24,25,26]. The synthesis of MT induced by low-dose zinc, copper, mercury, or cadmium can reduce the mortality of experimental mice caused by high-dose cadmium poisoning [27]. Transgenic mice overexpressing metallothionein genes showed a significant decrease in mortality due to cadmium poisoning, while knocking out the MT gene resulted in the opposite result [28]. The overexpression of *BSFMT1*, *BSFMT2A*, and *BSFMT2B* was found to confer Cd tolerance in *Escherichia coli* (*E. coli*) [29]. MT can not only relieve the toxicity of heavy metals but can also maintain the dynamic balance of trace elements and scavenge free radicals in vivo. The differences in Cd kinetics among *MT-I* transgenic and *MT-I* and *-II* knock-out animals and WT mice showed that MT reduces the elimination of Cd from the liver, and the knock-out of the *MT* gene also renders animals/cells more vulnerable to oxidative stress [30,31]. *MT-1* and *MT-2* activated by NO and peroxide confer neuroprotection in focal cerebral ischemia mouse models by upregulating *MTF-1* expression in the nucleus [32].

While most mammals show a single *MT1* and *MT2* copy which evolved through a duplication event, humans have multiple *MT1* copies [33,34]. In humans, *MT1* expansion resulted in a total of 18 tandemly arranged genes, 8 of which have been determined to be active genes (*MT1A*, *MT1B*, *MT1E*, *MT1F*, *MT1G*, *MT1H*, *MT1M*, *MT1X*) [26,35]. The *HsMT1L* gene belongs to the mammalian *MT1* subfamily and is expressed widely in human tissues [36,37]. It has been reported that the *HsMT1L* gene expression was upregulated 49.8-fold when exposed to zinc in vitro [38]. However, the biological function of *MT1L* is not fully understood. In this study, we synthesized the *HsMT1L* gene sequence and introduced it into the tobacco genome under the control of the CaMV 35S promoter to investigate the enhancement of the zinc and cadmium tolerance and enrichment of transgenic plants.

## 2. Materials and Methods

### 2.1. Experimental Materials and Reagents

Wild species of bare-stem tobacco (*Nicotiana nudicaulis* Watson, 2n = 2x = 24) was provided by our laboratory. *E. coli* str. Top10 and *Agrobacterium tumefaciens* LBA4404 was purchased from Beijing China Dingguo Biotechnology Co., Ltd. PCR Primers were synthesized by Shenzhen China Huada Biotechnology Co., Ltd. Taq DNA polymerase and DNA Mark were purchased from TaKaRa company; RNA extraction reagent was purchased from Dalian Bao Biological Company; real Time PCR Kit was purchased from Bio-Rad Biological Company; reverse transcription kit, gel recovery kit and plasmid extraction kit were purchased from Tiangen and Omega, respectively; MDA content, hydrogen peroxide content, superoxide dismutase, peroxidase, and catalase activities were all purchased from Suzhou China Keming Co., Ltd. Kit; kanamycin (Kan), Streptomycin (Str), Rifampicin (Rif), Zeatin (ZT) and other antibiotics and hormones were purchased from Beijing Dingguo Changsheng Biotechnology Co., Ltd.

### 2.2. Generation of Tobacco Plants Expressing HsMT1L

The human metallothionein gene *HsMT1L* (GenBank accession number: X76717) was synthesized by Sangon Biotech (Shanghai) Co., Ltd. It was controlled by the CaMV 35S promoter and was transformed into *N. nudicaulis* via Agrobacterium-mediated transformation [39,40].

The total RNA from the tobacco leaves, isolated with RNAiso Plus (TaKaRa, Dalian, China) as described in the manufacturer’s instructions, was treated with DNase I (TaKaRa, Dalian, China). The DNase-treated RNA was reverse-transcribed using a PrimeScript^TM^ RT Reagent Kit (TaKaRa, Dalian, China). qRT-PCR (quantitative real-time PCR) was performed using a CFX96 Real-Time PCR System (Bio-Rad Laboratories, Hercules, CA, USA) with Eva Green S (Bio-Rad Laboratories, Hercules, CA, USA). The relative expression of the detected gene was calculated using the 2^−∆∆Ct^ method.

Three highly expressed lines were screened by qRT-PCR using the primer pair PCL-forward (5′-GTGAGCGGATAACAATTCCC-3′) and PCL-reverse (5′-CAGAGAGACCGGATATAGTTCCTC-3′), and the Nb 18s rRNA gene was used as an internal control for normalization using the primer pair Nb 18s rRNA-forward (5′-AGTCTTTCGCTTTCTCACCATCTGCT-3′) and Nb 18s rRNA-reverse (5′-CTGCAAGAATCTCAAACACG -3′). Homozygous lines were obtained by PCR (polymerase chain reaction) from the T_2_ generation of the three highly expressed lines

### 2.3. Growth Conditions and Treatments

Weigh sterilized and air-dried soil (3 kg each plate), mix with 5 g compound fertilizer and 3 g K_2_SO_4_ as base fertilizer, and then add different concentrations of heavy metals. The ZnSO_4_ concentrations were 0, 300, and 900 mg/kg, and the CdCl_2_ concentrations were 0, 100, and 200 mg/kg. When the wild-type (WT) and transgenic plants were grown to 4–5 true leaves, seedlings of similar size were selected and transplanted into the plate for three weeks.

### 2.4. Determination of Zinc and Cadmium in Plants

The sample was decomposed using the ashing method. Wash 2 g fresh leaves and 1 g roots, dry the surface moisture, and grind into homogenate, respectively. Put in the pan and evaporate in the electric drying oven (pay attention to the temperature at 105–120 °C to prevent splash). The dried samples were carbonized in an electric furnace (105-120 °C) until the contents turned black and the smoke stopped. After carbonization, the sample was transferred into the porcelain crucible and burned at 525 ± 25 °C for about 8 h in the muffle furnace. White or gray residue indicates that the ashing is complete. The ceramic crucible was taken out and cooled, and 2.0 mL of 0.5 mol/L NaOH was slowly added along the wall to dissolve the ash. The solution was transferred into a 50 mL volumetric flask, and the ceramic crucible was washed with 30 mL of sodium hydroxide solution several times. The solution was poured into the same volumetric flask and diluted with distilled water to be measured. Three parallel tests for the same sample were conducted simultaneously with 10 mL distilled water as a blank reagent. The contents of zinc and cadmium were calculated according to the standard curve and formula based on the readings on the TAS-990 atomic absorption spectrophotometer.

### 2.5. Measurement of Antioxidant Enzyme Activity

To measure the antioxidant enzyme activity, 0.1 g of fresh material was weighed and put into the bowl, 500 μL of 50 mmol/L phosphate buffer (pH = 7.0, containing 1 mM EDTA-Na_2_ and 2% *w*/*v* PVP) was added to grind the homogenate, then centrifuged the homogenate at 4 °C and 11,000 rpm for 20 min. The supernatant enzyme solution was separated and loaded, and catalase (CAT), superoxide dismutase (SOD), and peroxidase (POD) activities were measured on the microplate reader (ice bath operation was used throughout the process). The POD activity was determined by the guaiacol oxidation method; the SOD activity was determined by the pyrogallol autoxidation method; the CAT activity was determined by the ammonium molybdate colorimetric method [41].

### 2.6. Determination of Chlorophyll Content in Plants

The 0.1 g of fresh material was loaded into the centrifuge tube, and 10 mL of the extract (acetone: ethanol = 1:1) was added. After thoroughly shaking, the leaves were kept in the dark for less than 24 h until the leaves were completely white. The absorbance values of the supernatant at 663 nm, 652 nm, and 645 nm were measured, and then the chlorophyll content was calculated by the formula [42].

### 2.7. Determination of Malondialdehyde Content in Plants

Malondialdehyde (MDA) was measured through the colorimetric method [43]. The thiobarbituric acid (TBA) reaction was used. The material was weighed at 0.3 g and loaded into the centrifuge tube. After adding 10% thiobarbituric acid (TCA) solution to 3 mL, the mixture was centrifuged at 4000 rpm for 10 min. Then 2 mL of the supernatant (control and 2 mL distilled water) were carefully absorbed with 3 mL 0.5% TBA solution and mixed in boiling water for 15 min. Afterwards, the material was rapidly cooled in ice and centrifuged at 4000 rpm for 10 min. The supernatant was taken to determine the absorbance values at 532 nm, 600 nm, and 450 nm, and the MDA content was calculated by the formula.

### 2.8. DAB Staining

Leaves of the WT and of transgenic plant line N14 were subjected in vitro to 200 mM/L CdCl_2_ or 300 mM/L ZnSO_4_ for 15 min, 0.5 h, 1 h, 3 h, or 6 h, respectively. Then the leaves were washed with distilled water and immersed in 1% 3,3′-diaminobenzidine (DAB) (50 mM Tris-HCl, pH 3.8) in the dark for 6 h. The leaves were then transferred to anhydrous ethanol and bathed in boiling water for about 10 min until the chlorophyll completely disappeared [44].

### 2.9. Subcellular Localization of HsMT1L

The *HsMT1L* gene was fused into *EGFP* and controlled by the CaMV 35S promotor. After obtaining the transgenic overexpression plants, fluorescence localization was observed using laser scanning confocal microscopy.

### 2.10. Statistical Analysis

All of the above tests were repeated three times independently. The measured data were statistically analyzed and plotted using Excel (version 2019) and GraphPad Prism software (GraphPad Software 8.0.2, San Diego, CA, USA).

## 3. Results

### 3.1. Expression of HsMT1L in Tobacco

To examine the biological function of *HsMT1L* in plants’ responses to heavy metal stress, we generated transgenic tobacco plants constitutively expressing *HsMT1L* under the control of the CaMV 35S promoter. The expression of the *HsMT1L* gene in different transgenic tobacco lines was analyzed by qRT-PCR, and several lines with high expression were screened for future experiments. The results (Figure 1) showed that the *HsMT1L* gene was not expressed in the WT but was expressed to different degrees in 12 transgenic tobacco plants, and the expression levels were quite different. Among them, the expression levels of the N13, N14, N21, and N24 lines were relatively high, five to six times higher than that of the N34 line, which had the lowest expression level, indicating that the different positions of the genes inserted into the plant chromosomes did affect the expression of the transgenic genes in the transgenic tobacco.

### 3.2. HsMT1L Gene Improved the Tolerance of Tobacco to Zn^2+^ and Cd^2+^

The 4-week-old WT and transgenic homozygous lines (N13, N14, N24) were selected to analyze the tolerance of Zn^2+^ and Cd^2+^. The plants were treated with Zn^2+^ at concentrations of 0, 300, or 900 mg/kg or Cd^2+^ at concentrations of 0, 100, or 200 mg/kg for three weeks to observe the growth status of tobacco. Under normal conditions or when treated with 100 mg/kg Cd^2+^, there was no apparent difference between the WT and the transgenic tobacco (Figure 2). When treated with Zn^2+^ and higher concentrations of Cd^2+^, the WT tobacco had more serious etiolation and dead spots, and its growth was slower than that of the transgenic tobacco (Figure 2). The toxicity of Zn^2+^ mainly showed the etiolation of leaves, withered spots, and gradual blackening of roots. The Cd^2+^ toxicity manifested in weak seedlings, chlorosis, and a slow growth rate (Figure 2).

The fresh weight of the WT and transgenic homozygous tobacco grown under Zn^2+^ or Cd^2+^ for three weeks was analyzed. The results showed that the growth of the transgenic and WT plants displayed no noticeable differences under normal growth conditions. Under the conditions of 300 or 900 mg/kg of Zn^2+^, the fresh weight of the transgenic tobacco was 36.5−50.5% or 45.3−79.1% higher (N13, *p* < 0.05; N14 and N24, *p* < 0.01) than that of the wild type, respectively (Figure 3a). No significant weight differences were detected between the WT and any transgenic line under 100 mg/kg of Cd^2+^ (Figure 3b). The fresh weight of the transgenic plants was 89.8−100.1% higher (*p* < 0.01) than that of the WT under 200 mg/kg Cd^2+^.

Under normal growth conditions, there was no significant difference in chlorophyll content between the transgenic and WT plants. Under 300 or 900 mg/kg Zn^2+^, the chlorophyll content of the transgenic tobacco was 9.9−15.9% or 45.5−54.1% higher than that of the wild type, respectively (Figure 3c). Under 200 mg/kg of Cd^2+^, the chlorophyll content of the transgenic tobacco was 5.8−14.7% higher than that of the WT, while it showed no obvious difference under the 100 mg/kg of Cd^2+^ treatment (Figure 3d). Among the above treatment concentrations, the difference was significant only for the 900 mg/kg of Zn^2+^ treatment (N13, *p* < 0.05; N14 and N24, *p* < 0.01). These results indicated that the heterologous expression of the *HsMT1L* gene alleviated the adverse effects of Zn^2+^ and Cd^2+^ stress on tobacco growth.

**Figure 3 genes-13-02413-f003:**
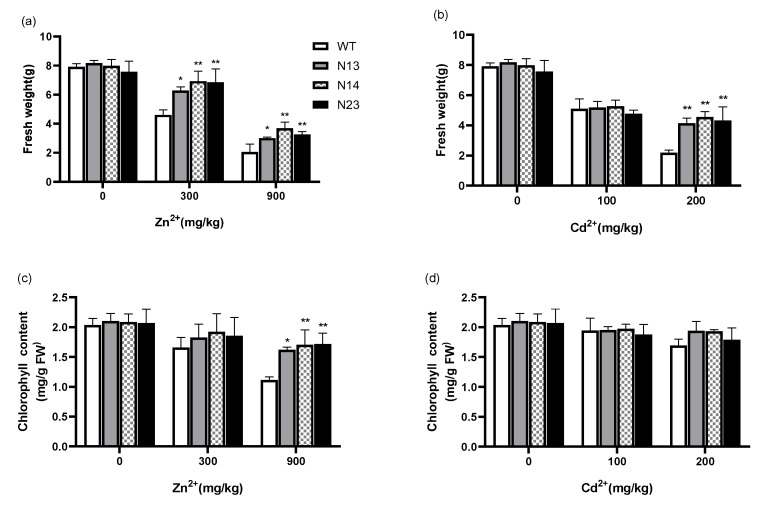
The fresh weight and chlorophyll content under Zn^2+^ or Cd^2+^ stress. (**a**) Fresh weight of the WT and transgenic lines under Zn^2+^ at different concentrations. (**b**) Fresh weight of the WT and transgenic lines under Cd^2+^ stress at different concentrations. (**c**) Chlorophyll content of the WT and transgenic lines under Zn^2+^ Cd^2+^ stress at different concentrations. (**d**) Chlorophyll content of the WT and transgenic lines Cd^2+^ stress at different concentrations. WT: Wild-type tobacco. N13, N14, N24: Three homozygous transgenic tobacco lines (T_2_). The 4−week−old WT and transgenic plants were treated under Zn^2+^ (0, 300, or 900 mg/kg) or Cd^2+^ (0, 100, or 200 mg/kg) for 3 weeks, respectively. Asterisks indicate a significant difference between transgenic lines and WT (*, *p* < 0.05; **, *p* < 0.01).

### 3.3. Heterologous Expression of HsMT1L Increased the Accumulation of Zn^2+^ and Cd^2+^

MT plays a vital role in detoxification due to its ability to bind heavy metal ions [14,45]. To understand the effects of *HsMT1L* expression on the accumulation of Zn^2+^ and Cd^2+^ in transgenic plants, the contents of Zn^2+^ and Cd^2+^ in the shoots and roots of the tobacco were determined, respectively. Under normal growth conditions, the Zn^2+^ and Cd^2+^ contamination levels in the shoots and roots of the transgenic and WT plants were low and had no significant differences. Under 300 mg/kg of Zn^2+^, the accumulation in the aboveground part of the transgenic plants was 12.6−18.7% higher than that of the WT (Figure 4a), with only N13 reaching a significant level compared with the WT (*p* < 0.05), and the Zn^2+^ contamination in the roots of the transgenic tobacco was 14.2−21.2% higher (N13 and N24, *p* < 0.001; N14, *p* < 0.001) than that of the WT (Figure 4b). Under 900 mg/kg of Zn^2+^, the contamination in shoots and roots of all of the transgenic plants was 26.2−33.1% (*p* < 0.0001) and 12.3−22.9% (N13 and N14, *p* < 0.0001; N24, *p* < 0.01) higher than that of the WT, respectively (Figure 4a,b). Under 100 mg/kg of Cd^2+^, the Cd^2+^ content in the aboveground and underground parts of the transgenic tobacco was 11.1−16.6% (N13, *p* < 0.001; N14 and N24, *p* < 0.05) and 24.3−26.5% (*p* < 0.001) higher than that of the WT, respectively (Figure 4c,d). Under 200 mg/kg of Cd^2+^, the Cd^2+^ contamination in the shoots and roots of transgenic plants was 24.6−30.5% and 45.5−51.3% higher than that of the WT, respectively (Figure 4c,d), and all of the transgenic lines showed a highly significant difference compared with the WT (*p* < 0.0001). The results showed that the transgenic plants had a high capacity to chelate Cd^2+^ and Zn^2+^. The higher the treatment concentration, the more significant the accumulation of Zn^2+^ and Cd^2+^ in transgenic plants.

### 3.4. HsMT1L Enhances the Antioxidant Capacity of Transgenic Tobacco under Zn^2+^ or Cd^2+^ Stress

Heavy metals can damage the physiological and biochemical functions of plant cells, leading to the formation of ROS [46]. An increasing ROS concentration poses the threat of oxidative stress to plant cells and may lead to lipid peroxidation [47]. MDA is the end product of cell membrane lipid peroxidation. Its content can be used as an essential physiological index of plant membrane lipid peroxidation damage under heavy metal and metal stress [48]. It can also reflect the strength of a plant’s stress resistance. The results showed no significant difference in the MDA content between the transgenic and the WT tobacco under normal growth conditions. Under 300 or 900 mg/kg of Zn^2+^, the MDA content of the transgenic tobacco was 67.3−70.7% or 58.1−66.4% that of the wild type, respectively (Figure 5a), and only under 900 mg/kg of Zn^2+^ were the differences between the transgenic lines and the WT significant (*p* < 0.01). Under 100 mg/kg of Cd^2+^, the MDA content of N13 and N14 was significantly less (17.3% and 23.1%, *p* < 0.05) than that of the WT (Figure 5b). Although the MDA content of N24 was less than that of the WT, the difference was insignificant (Figure 5b). Under 200 mg/kg of Cd^2+^, the MDA content of the transgenic lines was significantly less (53.3−58.6%, *p* < 0.001) than that of the WT (Figure 5b). The results showed that the heterologous expression of the *HsMT1L* gene could reduce the membrane peroxidation of transgenic plants.

CAT, SOD, and POD are the key enzymes that eliminate the excess of ROS accumulated under heavy metal stress [49]. These antioxidant enzymes can maintain the balance of reactive oxygen metabolism in cells to protect plant cells from oxidative damage [50]. The results showed no significant difference in antioxidant enzyme activity between the transgenic and WT plants under normal growth conditions.

Under the conditions of 300 mg/kg of Zn^2+^ or 900 mg/kg of Zn^2+^, the CAT activity of the transgenic tobacco was 59.7−71.9% (*p* < 0.0001) or 43.2−54.7% (N13 and N14, *p* < 0.001; N24, *p* < 0.0001) higher than that of the WT, respectively (Figure 6a). Under 200 mg/kg of Cd^2+^, the CAT activity of the transgenic tobacco was 35.2−46.4% higher (N13 and N14, *p* < 0.01; N24, *p* < 0.001) than that of the WT (Figure 6b). Under 100 mg/kg of Cd^2+^ conditions, the measurements showed no significant difference in CAT and SOD activity between the transgenic and WT plants. Under 300 or 900 mg/kg of Zn^2+^, the SOD activity of the transgenic tobacco was 66.9−76.1% (N13 and N24, *p* < 0.01; N14, *p* < 0.001) or 45.7−57.3% (*p* < 0.01) higher than that of the WT, respectively (Figure 6c). Under 200 mg/kg of Cd^2+^, the SOD activity of the transgenic tobacco was 36.1−59.9% higher (N14, *p* < 0.05; N24, *p* < 0.01) than that of the WT (Figure 6d). Under the conditions of 300 mg/kg or 900 mg/kg of Zn^2+^, the POD activity of the transgenic tobacco was 79.5−97.4% (N14 and N24 *p* < 0.01; N14, *p* < 0.001) or 23.6−50.7% (N14, *p* < 0.05; N24, *p* < 0.01; N14, *p* < 0.001) higher than that of the WT, respectively (Figure 6e). Under the conditions of 100 mg/kg of Cd^2+^, the POD activity of transgenic tobacco was 12.9−32.7% higher than that of the WT (Figure 6f), and only in N13 did it reach a significant level (*p* < 0.05). Under 200 mg/kg of Cd^2+^, the POD activity of the transgenic tobacco was 55.9−78.6% (N24, *p* < 0.01; N13 and N14, *p* < 0.001) higher than that of the WT (Figure 6f). These results suggested that the heterologous expression of the *HsMT1L* gene increased the activities of CAT, SOD, and POD in tobacco under Zn^2+^ or Cd^2+^ stress.

The released oxygen H_2_O_2_ can react with DAB to form a brown precipitate [51]. To study the oxidative stress level in tobacco leaves under heavy metal stress, we can analyze the degree of H_2_O_2_ damage on tobacco leaves by observing the histochemical staining of DAB. Leaves of the WT and of transgenic plant line N14 were subjected in vitro to 200 mM/L of CdCl_2_ or 300 mM/L of ZnSO_4_ for 15 min, 0.5 h, 1 h, 3 h, or 6 h, respectively. DAB staining was performed after 15 min, 30 min, 1 h, 3 h, and 6 h treatments, respectively. The results showed no significant differences between the WT and N14 leaves after the 15 min treatments (Figure 7). The area of color spots on leaves increased and the color deepened with time in each treatment of 0.5–6 h (Figure 7). Still, the degree of leaf discoloration of the transgenic tobacco was significantly lower than that of the WT tobacco (Figure 7). These results indicated that the accumulation of H_2_O_2_ reactive oxygen species in the transgenic leaves was less than that in the WT tobacco leaves. The results showed that the *HsMT1L* gene reduced the content of H_2_O_2_ in tobacco leaves under Cd^2+^, Zn^2+^, and H_2_O_2_ stress.

### 3.5. Subcellular Localization of HsMT1L in Transgenic Tobacco

Transgenic tobacco with an overexpression of *35S::HsMT1L-EGFP* was cultured, and the protein localization was observed under laser confocal microscopy. The microscopic observation showed that the green fluorescence in cells was observed both in the nucleus and the cytosol (Figure 8).

## 4. Discussion

Heavy metal stress can cause various damages to plants, mainly including enzyme inactivation, osmotic stress, and oxidative stress [52,53,54]. It is well-documented that MTs play a vital role in heavy metal detoxification and ROS scavenging [55]. MTs are characterized by a high content of Cys residues, which can effectively bind various bivalent metal ions [20]. Previously, it was reported that Zn induced the *HsMT1L* gene expression in vitro [38]. In this study, we expressed *HsMT1L* under the control of the CaMV 35S promoter in transgenic tobacco plants to clarify its biological function.

Some studies have demonstrated that expressing different MT genes in plants could lead to heavy metal tolerance and accumulation. Overexpression of *CcMT1* in *Arabidopsis* resulted in a strong resistance to copper and cadmium [56]. The heterologous expression of *PpMT2* in transgenic *Arabidopsis* plants conferred a tolerance to high concentrations of CuSO_4_ and CdCl_2_ [57]. *SaMT2* could significantly enhance the Cd^2+^ and Zn^2+^ accumulation in transgenic tobacco plants by chelating metals and improving the antioxidant system [58]. Similar results were also presented in this study. We found that *HsMT1L* transgenic tobacco accumulated Cd and Zn to higher levels, notably in the roots. A possible reason for this distribution is that roots were directly exposed to the heavy metals, and most of the metal ions were bound to MT and fixed in roots. Rapid Cd and Zn chelation by MT in the roots might alleviate the impacts of Cd and Zn toxicity on plants. The transgenic tobacco plants exhibited a tolerance of tobacco to Zn^2+^ and Cd^2+^, characterized by their fresh weight and chlorophyll content.

As a heavy metal chelator and reactive oxygen species scavenger, MTs can effectively alleviate the harmful effects of ROS induced by heavy metals [59,60]. This study demonstrated that the overexpression of *HsMT1L* could significantly reduce the H_2_O_2_ and MDA accumulation in tobacco exposed to Cd or Zn. MDA is the final decomposition product of membrane lipid peroxidation, and the lower MDA level indicated less lipid peroxidation and membrane damage [61]. Various antioxidant enzymes in plants also serve as an essential antioxidant defense mechanism in plants, such as CAT, SOD, and POD [62]. The higher the antioxidant enzyme activities, the stronger the resistance of the plants [52]. This study demonstrated that the tobacco plants overexpressing *HsMT1L* showed higher antioxidant enzyme activities. These results indicated that the *HsMT1L* gene could reduce the membrane lipid peroxidation and enhance the antioxidant capacity, thus improving tolerance to Zn^2+^ and Cd^2+^.

MTs are evolutionarily conserved and cysteine-rich proteins observed in the nucleus or cytoplasm of many cell types and tissues [63,64]. MT was localized in the nucleus and the cytoplasm of hepatocytes in newborn rats [65]. The OsMT2b and OsMT2c fused at both the N- and C-terminal of GFPs were all localized to the cytosol and nucleus [66]. Similar result was observed in this study. We found that HsMT1L was also localized to nucleus and cytosol in transgenic tobacco. Studies showed that proteins localized in both the nucleus and cytoplasm have 9.75 interaction partners on average [67]. The subcellular localization provides a clue for the identification of the function of HsMT1L.

## 5. Conclusions

In this study, the human *MT1L* (*HsMT1L*) gene under the control of the CaMV 35S promoter was transformed into tobacco. The accumulation of Zn^2+^ or Cd^2+^ in the transgenic tobacco plants presented as significantly higher than that in the WT, and the heterologous expression of HsMT1L enhanced the tobacco’s tolerance to Zn^2+^ and Cd^2+^.

## Figures and Tables

**Figure 1 genes-13-02413-f001:**
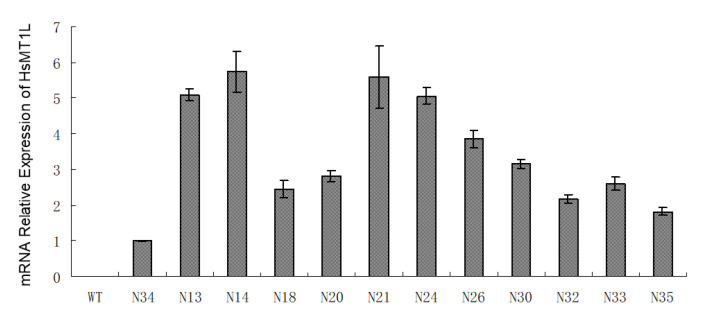
*HsMT1L* transcript levels among different lines. WT: wild-type tobacco; N13~N35: *HsMT1L* transgenic lines.

**Figure 2 genes-13-02413-f002:**
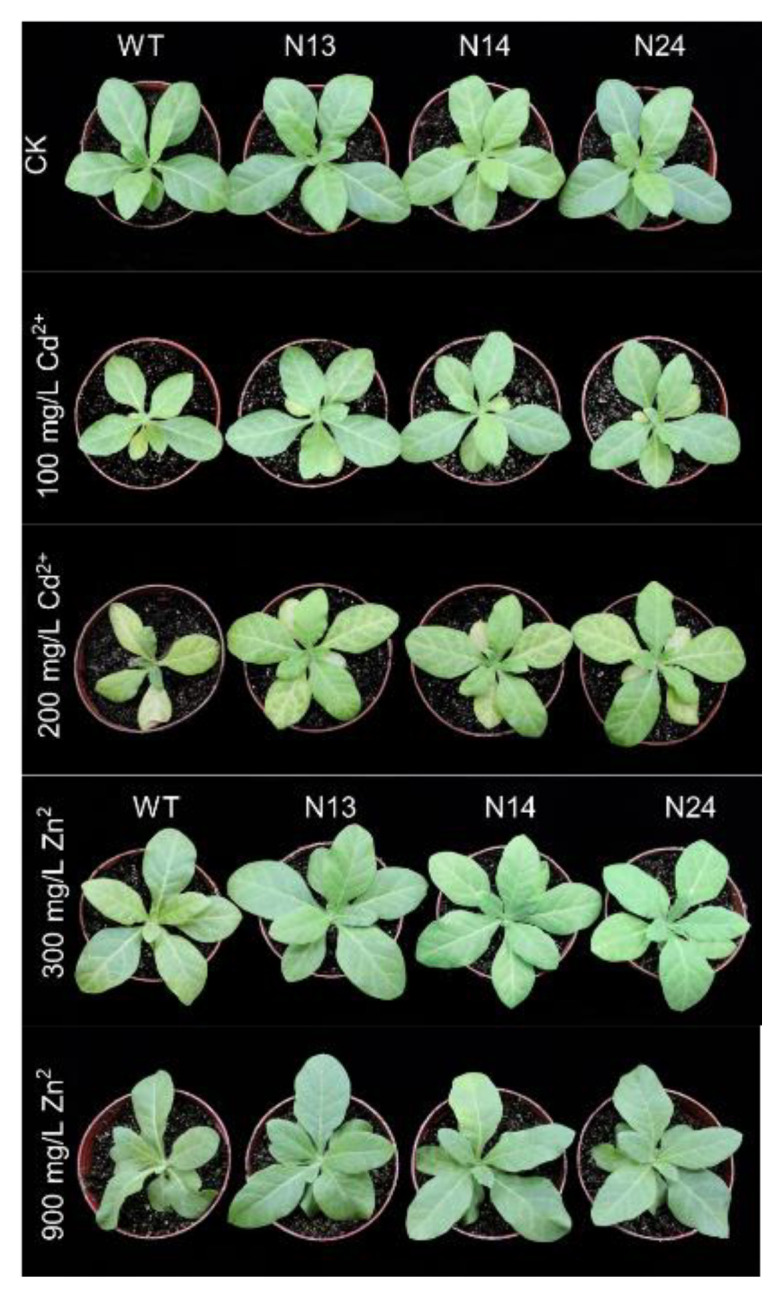
Tolerance of transgenic and WT plants to Cd^2+^ or Zn^2+^ stress. WT: Wild−type tobacco. N13, N14, N24: Three homozygous transgenic tobacco lines (T_2_). Four−week−old transgenic lines and WT plants were grown under Zn^2+^ (0, 300, or 900 mg/kg) or Cd^2+^ (0, 100, or 200 mg/kg) for 3 weeks, respectively.

**Figure 4 genes-13-02413-f004:**
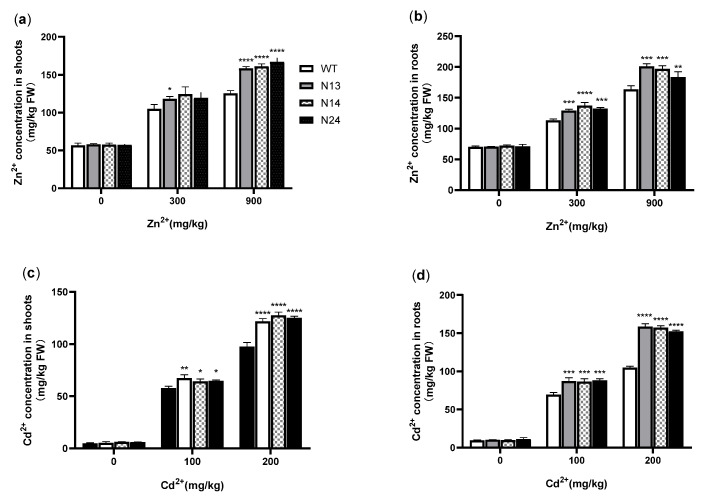
Zn^2+^ and Cd^2+^ concentrations of transgenic and WT tobacco in shoots and roots. (**a**) Zn contents in shoots. (**b**) Zn contents in roots. (**c**) Cd contents in shoots. (**d**) Cd contents in roots. WT: Wild-type tobacco. N13, N14, N24: Three homozygous transgenic tobacco lines (T_2_). The 4−week−old WT and transgenic plants were treated under Zn^2+^ (0, 300, or 900 mg/kg) or Cd^2+^ (0, 100, or 200 mg/kg) for 3 weeks, respectively. Asterisks indicate the significant difference between transgenic lines and WT (*, *p* < 0.05; **, *p* < 0.01; ***, *p* < 0.001; ****, *p* < 0.0001).

**Figure 5 genes-13-02413-f005:**
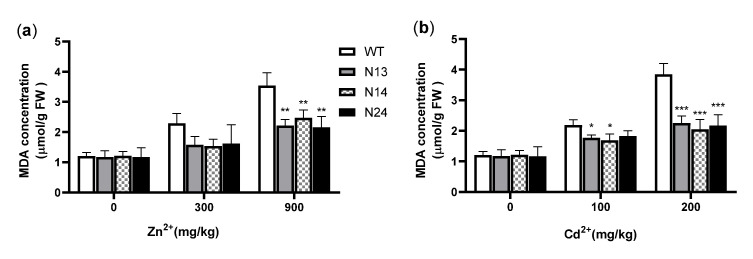
Comparison of MDA content under Zn^2+^ or Cd^2+^ stress. (**a**) MDA content of transgenic and WT tobacco under different concentrations of Zn^2+^. (**b**) MDA content of transgenic and WT tobacco under different concentrations of Cd^2+^. WT: Wild-type tobacco. N13, N14, N24: Three homozygous transgenic tobacco lines (T_2_). The 4−week−old WT and transgenic plants were treated under Zn^2+^ (0, 300, or 900 mg/kg) or Cd^2+^ (0, 100, or 200 mg/kg) for 3 weeks, respectively. Asterisks indicate the significant difference between transgenic lines and WT (*, *p* < 0.05; **, *p* < 0.01; ***, *p* < 0.001).

**Figure 6 genes-13-02413-f006:**
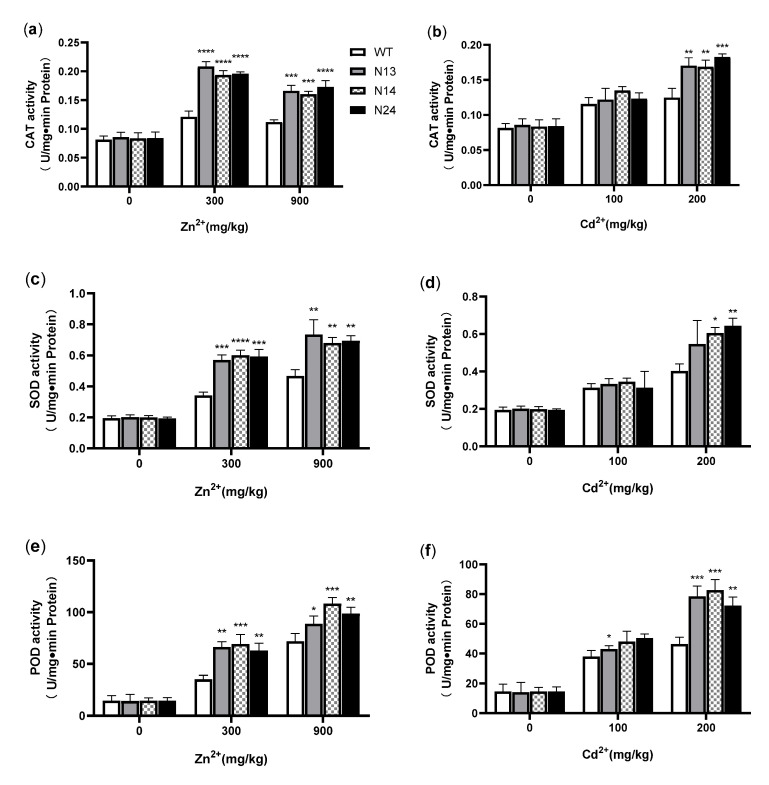
Comparison of antioxidant enzyme activities under Zn^2+^ or Cd^2+^ stress. (**a**) CAT activity of transgenic and WT tobacco under different concentrations of Cd^2+^. (**b**) CAT activity of transgenic and WT tobacco under different concentrations of Zn^2+^. (**c**) SOD activity of transgenic and WT tobacco under different concentrations of Zn^2+^. (**d**) SOD activity of transgenic and WT tobacco under different concentrations of Cd^2+^. (**e**) POD activity of transgenic and WT tobacco under different concentrations of Zn^2+^. (**f**) POD activity of transgenic and WT tobacco under different concentrations of Cd^2+^ stress. WT: Wild-type tobacco. N13, N14, N24: Three homozygous transgenic tobacco lines (T_2_). The 4−week−old WT and transgenic plants were treated under Zn^2+^ (0, 300, or 900 mg/kg) or Cd^2+^ (0, 100, or 200 mg/kg) for 3 weeks, respectively. Asterisks indicate the significant difference between transgenic lines and WT (*, *p* < 0.05; **, *p* < 0.01; ***, *p* < 0.001; ****, *p* < 0.0001).

**Figure 7 genes-13-02413-f007:**
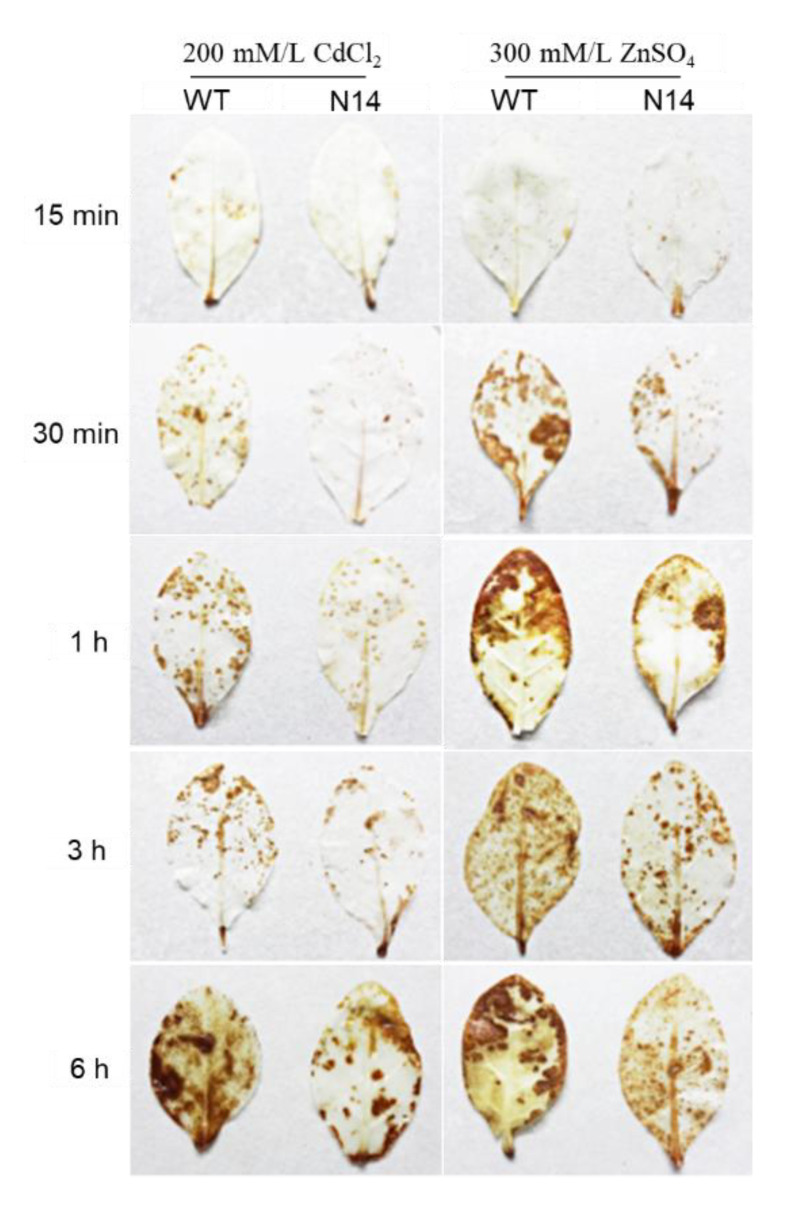
Production of hydrogen peroxide detected by DAB staining under Zn^2+^ or Cd^2+^ stress. DAB staining of WT and N14 leaves subjected to 200 mM/L CdCl_2_, 300 mM/L ZnSO_4,_ or 30 mM/L H_2_O_2_ for 15 min, 30 min, 1 h, 3 h, or 6 h, respectively. WT: Wild-type tobacco. N14: homozygous transgenic tobacco line (T_2_). Asterisks indicate the significant difference between transgenic lines and WT.

**Figure 8 genes-13-02413-f008:**
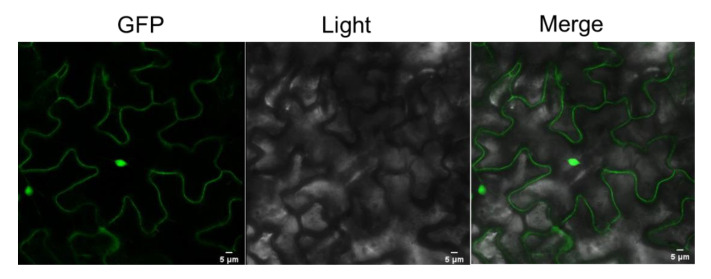
Subcellular localization of HsMT1L. Images were taken by using Leica confocal microscopy. GFP: GFP fluorescence under green light; Light: visible light image; Merge: merge images of the above two images.

## Data Availability

Not applicable.

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
