# Peer review of "Heterologous Expression of Human Metallothionein Gene HsMT1L Can Enhance the Tolerance of Tobacco (Nicotiana nudicaulis Watson) to Zinc and Cadmium"

_genes, 2022, doi:10.3390/genes13122413_

Round 1

Reviewer 1 Report

The article deals with a very systematic and comprehensive study on the transformation of tobacco with the human metallothionein gene HsMT1L and testing of T2 transgenic lines for expression and cellular localization of transgene, accumulation of zinc and cadmium in various plant parts  and tolerance to the heavy metals  at various concentrations using different  parameters in some of the transgenic lines with high level of expression of the HsMT1L gene. The results are well presented and explained. The Material & Methods section has to be rewritten using proper scientific English as suggested and high lighted using track changes in the reviewed manuscript attached. It will be better of some of the results depicted in figures 4-6 are given in a single table for better comparison of wild type and transgenic lines for  heavy metal accumulation and tolerance parameters at different concentrations. The result section also needs to be rewritten as suggested. The extent  and global distribution of zinc and cadmium toxicity should be given in a map to show the justification of such a study. Cadmium is not a micronutrient but there is wide spread deficiency of zinc  through out the world and it is usually applied and sprayed for proper crop production. There are concerted efforts by HarvestPlus and many countries to biofortify cereals for high grain zinc to alleviate wide spread zinc deficiency in many developing countries. It will be interesting to find if the metallothionein transgenic  plants have also higher zinc in grains under normal soil concentration compared to the non-transgenic lines. 

Author Response

Point: The article deals with a very systematic and comprehensive study on the transformation of tobacco with the human metallothionein gene HsMT1L and testing of T2 transgenic lines for expression and cellular localization of transgene, accumulation of zinc and cadmium in various plant parts and tolerance to the heavy metals at various concentrations using different parameters in some of the transgenic lines with high level of expression of the HsMT1L gene. The results are well presented and explained. The Material & Methods section has to be rewritten using proper scientific English as suggested and high lighted using track changes in the reviewed manuscript attached. It will be better of some of the results depicted in figures 4-6 are given in a single table for better comparison of wild type and transgenic lines for heavy metal accumulation and tolerance parameters at different concentrations. The result section also needs to be rewritten as suggested. The extent and global distribution of zinc and cadmium toxicity should be given in a map to show the justification of such a study. Cadmium is not a micronutrient but there is wide spread deficiency of zinc throughout the world and it is usually applied and sprayed for proper crop production. There are concerted efforts by HarvestPlus and many countries to biofortify cereals for high grain zinc to alleviate wide spread zinc deficiency in many developing countries. It will be interesting to find if the metallothionein transgenic plants have also higher zinc in grains under normal soil concentration compared to the non-transgenic lines.

Response: Thanks for the reviewer’s insightful and helpful comments. After considering these important suggestions and revising the manuscript accordingly. We have tried to change figure 6 to a single table, but we found that the figures seem clearer and present the results visually. Therefore, we hope we can keep these figures. We agree that a global distribution map is better to show the justification. We have read relevant studies about Cd and Zn pollution in some countries and regions, and we have addressed them in line29-30. We hope this will be convincing enough. We also think this is an interesting idea to find if transgenic plants can improve Zn concentration in grains. In our unpublished study, we demonstrated that overexpressing HsMT1L in tomatoes improved the Zn content in the fruit. We think HsMT1L might be a potential gene for Zn supplement food cultivation.

Reviewer 2 Report

It is a well-rounded presentation of the results. One of my main concerns is conclusions. The conclusions might be elaborate to highlight the lacunae in our present understanding of the mechanism regulating functions of metallothionein. Giving a guiding light for future scientists to continue the work in a similar direction for crop improvement. 

Author Response

Point: It is a well-rounded presentation of the results. One of my main concerns is conclusions. The conclusions might be elaborate to highlight the lacunae in our present understanding of the mechanism regulating functions of metallothionein. Giving a guiding light for future scientists to continue the work in a similar direction for crop improvement.

Response: We find these comments and the English revision very insightful . After considering these important suggestions and revising the manuscript accordingly, we hope the revisions are satisfactory.

Reviewer 3 Report

This study presents the human metallothionein gene HsMT1L's tolerance and accumulation capacity of transgenic tobacco plants to Zn and Cd. All experiments are correctly done, and the results are presented clearly. There are only a few minor mistakes to be taken into consideration before the publication of this study, as follows:

-There is no Figure a and b. 

-Figure 3d is cited before Figure 3c in the text. Please cite them in order. 

-Figure 4c is cited before Figures b and c in the text. Better to cite them in order.

-Line 336, 391, 520, 541, 553, Write Arabidopsis in italics. 

- Scientific species name of tobacco is mentioned only in the material and Methods. Please write the species name in the title as tobacco (Nicotiana nudicaulis Watson) if this is the species you studied. 

Author Response

Point 1: There is no Figure a and b

Response 1: Thank you for pointing this out. We have fixed the mistake.

Point 2: Figure 3d is cited before Figure 3c in the text. Please cite them in order.

Response 2: Thank you for the comment. We have modified the cite order.

Point 3: Figure 4c is cited before Figures b and c in the text. Better to cite them in order.

Response 3: Thank you for the comment. We have modified the cite order.

Point 4: Line 336, 391, 520, 541, 553, Write Arabidopsis in italics.

Response 4: Thank you for pointing this out. We have modified the word.

Point 5: Scientific species name of tobacco is mentioned only in the material and Methods. Please write the species name in the title as tobacco (Nicotiana nudicaulis Watson) if this is the species you studied.

Response 5: Thank you for the suggestion. We have modified the title.